# Photocatalytic Synthesis of Coumarin Derivatives Using Visible-Light-Responsive Strawberry Dye-Sensitized Titanium Dioxide Nanoparticles

**DOI:** 10.3390/nano13233001

**Published:** 2023-11-22

**Authors:** Mshari A. Alotaibi, Abdulrahman I. Alharthi, Talal F. Qahtan, Satam Alotibi, Amani M. Alansi, Md. Afroz Bakht

**Affiliations:** 1Chemistry Department, College of Science and Humanities, Prince Sattam Bin Abdulaziz University, P.O. Box 83, Al-Kharj 11942, Saudi Arabia; a.alharthi@psau.edu.sa (A.I.A.); m.bakht@psau.edu.sa (M.A.B.); 2Department of Physics, College of Science and Humanities, Prince Sattam Bin Abdulaziz University, P.O. Box 173, Al-Kharj 11942, Saudi Arabia; t.qahtan@psau.edu.sa (T.F.Q.); sf.alotibi@psau.edu.sa (S.A.); 3Chemistry Department, King Saud University, P.O. Box 2455, Riyadh 12372, Saudi Arabia; amanialansi92@gmail.com

**Keywords:** photocatalytic synthesis, coumarin derivatives, TiO_2_, strawberry dye, visible light, organic synthesis, green photochemistry

## Abstract

This study presents a novel method for the photocatalytic synthesis of 4-aryl-6-(3-coumarinyl) pyrimidin-2 (1H)-ones (a coumarin derivative) using strawberry dye-sensitized TiO_2_ (SD-TiO_2_) under visible light. The synthesis of 4-aryl-6-(3-coumarinyl) pyrimidin-2 (1H)-ones was achieved through a three-component, one-pot condensation reaction involving 3-acetyl coumarin, aldehydes, and urea, utilizing SD-TiO_2_ as a reusable and innovative photocatalyst at room temperature. The resulting SD-TiO_2_ photocatalyst was thoroughly characterized using FT-IR, XPS, XRD, SEM, and BET. The efficacy of SD-TiO_2_ was evaluated by comparing it to pristine TiO_2_ in terms of photocatalytic activity, and the optimal conditions for the synthesis process were determined. Notably, the SD-TiO_2_ photocatalyst exhibited a maximum yield of the compound, reaching up to 96% in just 30 min with a catalyst concentration of 1 mg/mL. This yield surpasses traditional thermal procedures employing reflux conditions, where 1 mg/mL of SD-TiO_2_ is sufficient to complete the reaction. The resulting 4-aryl-6-(3-coumarinyl) pyrimidin-2 (1H)-ones were further characterized using ^1^H-NMR and ^13^C-NMR. Moreover, the stability of the SD-TiO_2_ photocatalyst was confirmed through recyclability experiments and spectroscopic characterization, demonstrating its practicality for up to three consecutive reaction cycles.

## 1. Introduction

Multicomponent reactions (MCRs) possess many significant advantages, such as their simplicity, ability to save energy and time, high efficiency in bond formation, and low consumption. MCRs are used to synthesize coumarin derivatives because they play a crucial role in organic synthesis by facilitating the formation of carbon–carbon and carbon–heteroatom bonds in a single process [1,2,3]. Coumarin derivatives are important classes of heterocycles that are broadly applied across various industries, including pharmaceuticals, fragrances, and food, due to their diverse biological activities and appealing scents [4,5,6,7]. Due to their unique photophysical properties, coumarin derivatives show potential applications in organic solar cells, photovoltaic devices, and other renewable energy technologies [8,9,10,11]. These applications significantly contribute to the development of sustainable and efficient energy systems, aligning with the global agenda of transitioning towards cleaner and renewable sources of energy.

Traditional methods such as Pechmann condensation [12], the Perkin reaction [13], the Knoevenagel reaction [14], and microwave-assisted synthesis [15] have been employed for the synthesis of coumarin derivatives. However, these methods are often associated with harsh conditions, the use of toxic reagents, and multiple synthetic steps, which limit their application and pose significant environmental concerns. Consequently, there is a growing demand for more sustainable and more efficient approaches to the production of these compounds.

In recent times, chemists have become increasingly aware of the environmental impact of their chemical practices. Consequently, they are now striving to explore novel synthetic methods by choosing reaction conditions and chemicals which minimize risks to both humans and the environment [16]. In that regard, the increasing demand for sustainable alternatives to traditional synthetic methods has led to a growing interest in the field of photocatalysis [17,18]. The use of photocatalysis, a process where light energy triggers a chemical reaction through the activation of a photocatalyst, has been extensively researched due to its potential for sustainable organic synthesis. Photocatalysis under visible light has emerged as a promising method due to its efficiency and sustainability [19]. This process involves the activation of a photocatalyst by visible light, which subsequently promotes a chemical reaction. A key advantage of using visible light is that it constitutes a significant portion of the solar spectrum, thereby supporting the development of solar-driven chemical processes and the broader sustainability agenda [20,21].

Photocatalytic synthesis under visible light may represent a sustainable and efficient approach to produce coumarin derivatives by adjusting several parameters, including the type of photocatalyst, light intensity and wavelength, and reaction time; specific coumarin derivatives can be produced with high purity and yield [22]. Furthermore, this method eliminates the need for toxic reagents and solvents, ensuring safety for both operators and the environment [13,23,24].

An essential aspect of photocatalytic systems is the photocatalyst itself. Titanium dioxide (TiO_2_), a widely used photocatalyst, is often limited by its wide band gap energy, which restricts its use to only the ultraviolet region of the electromagnetic spectrum. One promising strategy to overcome this limitation involves sensitizing TiO_2_ with dyes that can absorb light in the visible region [16,25,26]. Utilizing dyes to sensitize TiO_2_ is a highly promising method for improving its photocatalytic activity in the presence of visible light. Many studies reported significant progress in developing dye-sensitized TiO_2_ photocatalysts, employing different dyes, including natural ones such as betalains, anthocyanins, and chlorophylls [27,28,29,30,31,32,33]. Nonetheless, further exploration of new natural dyes for TiO_2_ sensitization under visible light is warranted. Moreover, dye-sensitized TiO_2_ photocatalysts hold significant potential for energy applications, particularly in dye-sensitized solar cells [34]. Dye-sensitized solar cells offer a cost-effective alternative to traditional silicon solar cells, converting sunlight into electricity through dye molecules that absorb light and create an electric charge [34].

In the realm of environmental remediation, the utilization of dye-sensitized TiO_2_ has demonstrated remarkable potential in eliminating pollutants from both air and water [29,31]. One of the key advantages associated with employing dye-sensitized TiO_2_ in decontamination processes stems from its remarkable capability to harness solar energy and effectively convert it into chemical reactions that facilitate the breakdown of pollutants. This innovative approach has found practical applications in a wide range of decontamination scenarios, encompassing air purification systems, wastewater treatment, and the remediation of soil and groundwater that have been compromised by contamination [35,36,37]. Notably, this method presents a sustainable and environmentally conscious solution to decontamination challenges, as it heavily relies on renewable energy sources while avoiding the generation of harmful byproducts.

In this study, we successfully synthesize and characterize a strawberry dye-sensitized TiO_2_ (SD-TiO_2_) as a novel photocatalyst. We investigated the effect of the use of a natural dye extracted from strawberries to sensitize TiO_2_ nanoparticle photocatalysts for the photocatalytic synthesis of coumarin derivatives under visible light. Strawberry dyes are abundant in anthocyanins and are known for their strong absorption in the visible region of the electromagnetic spectrum [38]. The main aim was to assess the efficacy of SD-TiO_2_ as a sustainable and efficient tool for organic synthesis by comparing the photocatalytic activities of SD-TiO_2_ and pristine TiO_2_. The study confirmed that SD-TiO_2_ exhibited enhanced photocatalytic activity for the synthesis of coumarin derivatives under visible light. We investigated the optimal conditions for the photocatalytic synthesis process and proposed a plausible mechanism underlying the improved performance of the dye-sensitized photocatalyst. Additionally, we examined the reusability of SD-TiO_2_ over four consecutive runs to assess its practicality.

The integration of environmentally conscious practices into photocatalysis is crucial for sustainable organic synthesis. Using recyclable nanomaterials with light as catalysts offers a potential solution for atom-efficient transformations. However, there are challenges regarding the activation of metal oxide nanoparticles under UV light. This review article focuses on enhancing heterogeneous photocatalysts with dyes to overcome these limitations. It explores the design aspects and mechanisms of dye-sensitized photocatalysts, particularly in the synthesis of coumarin derivatives using visible light. This study represents a pioneering effort in the field of synthesizing coumarin derivatives using photocatalysts, making it the first of its kind. There are no similar studies available that use dye-sensitized TiO_2_ as a novel photocatalyst used for the photocatalytic synthesis of coumarin derivatives under visible light. It also underscores the potential of natural dyes to enhance the photocatalytic activity of TiO_2_, opening new avenues for research in the field of photochemistry. Overall, the field of photocatalytic synthesis of coumarin derivatives using visible light shows great promise and is poised to make significant contributions to various scientific and industrial applications.

## 2. Materials and Methods

### 2.1. Materials

For this study, the researchers procured P25 TiO_2_ nanoparticles from Evonik (Berlin, Germany). Chemicals, including dimedone, urea, thiourea, and various aldehydes such as 4-chlorobenzaldehyde, 4-fluorobenzaldehyde, 4-hydroxy-3-methoxybenzaldehyde (Vanillin), and furfural, were obtained from Sigma Aldrich in St. Louis, MI, USA.

### 2.2. Preparation of Strawberry Dye

We prepared the strawberry dye in ethanol as follows (refer to Figure 1a). First, we washed 100 g of fresh strawberries and removed any stems or leaves. Second, we cut the strawberries into small pieces and ground them via a mortar and pestle until a paste was formed. We transferred the strawberry paste to a glass beaker and added 100 mL of ethanol, ensuring the paste was fully submerged. After that, we thoroughly stirred the mixture via a magnetic stirrer to extract the color pigments from the strawberries into the ethanol. The beaker was covered with filter paper and left at room temperature for 24 h to allow the pigments to dissolve completely in the ethanol. To remove any solid particles or debris, we filtered the mixture through a funnel lined with filter paper. Finally, we transferred the filtered strawberry dye solution to an amber glass bottle to protect it from light and stored it in a cool, dark place until further use.

### 2.3. Preparation of Strawberry Dye-Sensitized TiO_2_

The sensitization of TiO_2_ nanoparticles with strawberry dye was achieved, as shown in Figure 1b,c, via the following steps. First, we incorporated 1 g of TiO_2_ nanoparticles into the filtered strawberry dye solution and then thoroughly stirred the mixture until the TiO_2_ nanoparticles were fully dispersed. To enhance the dispersion of TiO_2_ nanoparticles and promote their adhesion to the dye molecules, we placed the mixture into ultrasonic bath sonication for 30 min. To enable the adsorption of dye molecules onto the surface of TiO_2_ nanoparticles, we allowed the mixture to stand for 24 h. After that, we separated the TiO_2_ nanoparticles from any unbound dye molecules by centrifuging the mixture at 5000 rpm for 10 min. Then, we removed residual dye molecules or impurities by washing the TiO_2_ nanoparticles with deionized water.

Finally, the as-prepared strawberry dye-sensitized TiO_2_ nanoparticles could be used for the photocatalytic synthesis of coumarin derivatives under visible light.

### 2.4. Photocatalytic Synthesis of Coumarin Derivatives

The coumarin derivatives, as shown in Figure 1, were prepared by dissolving 0.5 mmol of 3-acetyl coumarin (1), 1 mmol of substituted aldehydes (2a–h), and 1 mmol of urea (3) in 10 mL of ethanol in a 50 mL glass beaker. The desired quantity of strawberry dye-sensitized TiO_2_ photocatalyst (1 mg/mL) was introduced into the mixture and thoroughly stirred for 30 min to ensure even dispersion of the photocatalyst. An ultrasonic bath was utilized to sonicate the mixture for 30 min, promoting further dispersion of the TiO_2_ nanoparticles and enhancing their adhesion to the reactant molecules. Under visible light irradiation, we employed a solar simulator equipped with a UV cut-off filter to expose the mixture for the desired duration.

After the photocatalytic reaction, the mixture underwent the addition of a suitable amount of deionized water using a funnel containing filter paper, aiming to eliminate any solid particles or debris. The product was then extracted using dichloromethane and the organic layer was separated. The organic layer was dried with anhydrous sodium sulfate (Na_2_SO_4_) and its purity was confirmed through thin layer chromatography (T.L.C) (Benzene: Acetone: 9:2). Additionally, NMR spectroscopy was employed to further verify the structure.

### 2.5. Characterization

To analyze the properties of pristine TiO_2_ and strawberry dye-sensitized TiO_2_, various analytical techniques were employed. FTIR spectroscopy was performed using the iD5 ATR diamond Nicolet instrument from Thermo Science (Waltham, MA, USA). UV-Vis spectrophotometry was conducted using Jasco’s V-570 instrument, manufactured by Jasco Corporation (Tokyo, Japan). To investigate the thermal properties, we conducted DTA/TG analysis using Netzsch Proteus 70 equipment, produced by Netzsch Gerätebau GmbH (Selb, Germany). The analysis was carried out by applying a heating rate of 10 °C/min within the temperature range of 25–1000 °C and utilizing an air environment. XRD analysis covered the scanning range of 2θ = 10−70° and was carried out using (Rigaku International, Tokyo, Japan) X-ray diffractometer with Cu Kα radiation (λ = 1.5406 A). SEM imaging was performed using FEI’s Quanta 250 instrument (SEM, QUANTA 250 FEI; Hillsboro, OR, USA). XPS analysis utilized a Thermo K Alpha spectrometer (XPS) Thermo Fisher Scientific (Waltham, MA, USA) operating at an energy value of 1486.6 eV with a spot size of 400 μm, and charge correction was applied during analysis. All binding energy estimates were calibrated to the C 1s energy (284.5 eV) as an internal standard. The BET surface area, pore radius, and pore volume of the catalyst were determined through N_2_-physisorption at 77 K using Quantachrome ASiQwin software version 5.2 (Boynton Beach, FL, USA). In the BET analysis for the SD-TiO_2_ sample, we placed the sample in a vacuum oven for a suitable degassing process and then introduced a gas (typically nitrogen) into the sample cell at a controlled pressure and temperature. After that, the photocatalytic activity of the prepared samples was investigated via continuous exposure to UV-visible light using a 500 W Xenon Light Source (TOP-X500, TOPTION, China) that functioned as a solar simulator. The 500 W Xenon Light Source consists of a housing (TSH-X500), power supply (TSP-X500), and Lamp (TSL-X500). To make use of the visible-light portion of the 500 W Xenon Toption and irradiate the photocatalyst samples, a UV cut-off filter was employed. The samples directly received visible light, with an average solar intensity measuring 100 mWcm^−2^. For a visual representation of the reaction setup and the lamp sources, please refer to the Appendix A, where a digital image is provided. Synthetic octahydroquinazolinones were analyzed using Bruker-Plus (400 MHz) nuclear magnetic resonance (NMR) equipment, thus obtaining the ^1^H-NMR and ^13^C spectra. Tetramethylsilane served as the internal reference during the spectral recording.

## 3. Results and Discussion

### 3.1. Photocatalysts Characterization

#### 3.1.1. FTIR Analysis

FTIR analysis was utilized to confirm the presence of strawberry dye in the prepared samples and to provide supporting evidence for their identification. The FTIR analysis effectively detected the presence of pristine TiO_2_, strawberry dye, and SD-TiO_2_ samples, as shown in Figure 2. Specifically, the FTIR spectrum of TiO_2_ (Figure 2a) exhibited three distinct bands. These bands included a broad and prominent one observed at 600–700 cm^−1^, representing the bending vibration (Ti-O-Ti) bonds within the TiO_2_ lattice. Another broad band centered at 3500–3000 cm^−1^ indicated the interaction between hydroxyl groups of water molecules and the TiO_2_ surface. Additionally, a weak peak at 1640 cm^−1^ corresponded to the characteristic bending vibration of the ^−^OH group [39,40,41].

Figure 2b displayed the FTIR spectrum of strawberry dye, which exhibited several characteristic absorption peaks. These peaks included absorptions at 3345.4 and 1651.2 cm^−1^, indicating the presence of -OH groups in the aliphatic and aromatic groups. The spectrum also revealed the presence of functional groups such as O-H, C-O alcohol, C-H aromatic, C-H aliphatic, and C-O ether functional groups, typically found in flavonoid compounds, confirming the existence of SB dye [42]. Intramolecular hydrogen bond vibrations were evident, supported by the vibration of C-O alcohol bending at 1044.2 cm^−1^, confirming the presence of OH groups attached to carbon atoms. The spectrum also indicated the absorption peak of the C-O ether group’s vibration and a strong absorption peak of the aromatic C-H bend at 1086.5 and 878.8 cm^−1^, respectively. Additionally, weak bands at 2973.7 and 2890.3 cm^−1^ indicated the presence of aliphatic C-H stretching vibrations.

The FTIR spectrum of SD-TiO_2_ (Figure 2c) exhibited new bands compared to the TiO_2_ nanoparticles spectrum, attributed to the presence of strawberry dye. Notably, these bands, including those around 2973.7 and 2890.3 cm^−1^, 1380.7 and 878.8 cm^−1^, and 1044.2 and 1086.3 cm^−1^, corresponding to C-H stretching vibrations, C-H bending, and C-O stretching vibration, respectively, showed a shift from their original positions in the strawberry dye spectrum. This shift indicated a hydrogen-bonding nature and confirmed the interaction between strawberry dye and the TiO_2_ surface. Furthermore, the absorption of Ti-O-Ti stretching vibrations for SD-TiO_2_ at 652.6 cm^−1^ was weaker compared to that of TiO_2_, providing support for the successful loading of strawberry dye onto the TiO_2_ surface.

By comparing the FTIR spectra, the functional groups present in the samples can be identified. The presence of characteristic peaks in the prepared SD-TiO_2_ spectrum, which are also observed in the strawberry dye spectrum but not in the spectrum of TiO_2_ alone, confirms the successful loading of strawberry dye onto pristine TiO_2_. This confirms the successful synthesis of SD-TiO_2_.

#### 3.1.2. XPS Analysis

X-ray photoelectron spectroscopy (XPS) is a technique renowned for its ability to analyze surface layers and provide valuable information about the chemical composition and electronic states of materials. Figure 3 illustrates the XPS analysis conducted on the TiO_2_ nanoparticles and SD-TiO_2_ samples. The XPS survey spectra of the pristine TiO_2_ and SD-TiO_2_ samples depicted in Figure 3a,b, respectively, confirm the presence of Ti, O, and C elements. The presence of a small C 1s peak located at 284.4 eV in pristine TiO_2_ is mainly attributed to its absorption from the atmosphere before XPS measurements. Moreover, the high-resolution Ti2p XPS spectra pertaining to pristine TiO_2_ and SD-TiO_2_ are displayed in Figure 3c,d. These spectra reveal two oxidation states of Ti: Ti ^3+^ and Ti ^4+^. Each oxidation state consists of two components, Ti 2p3/2 and Ti 2p1/2, with binding energies centered at 456.8 eV and 460.7 eV, respectively, for the Ti ^3+^ oxidation state and at 458.3 eV and 464.1 eV, respectively, for the Ti^4+^ oxidation state [43].

Figure 3e,f shows the XPS O 1s spectra of both pristine TiO_2_ and SD-TiO_2_, respectively. The O 1s spectrum of the pristine TiO_2_ (Figure 3e) comprises two components located at the binding energies 529.5 eV and 531.4 eV corresponding to the Ti-O-Ti and Ti-OH bonds, respectively [44]. Therefore, the XPS spectrum of pristine TiO_2_ confirms the presence of Ti and O elements in the nanoparticles. The XPS O 1s spectrum of SD-TiO_2_ (Figure 3f) exhibits four components representing different chemical environments of oxygen atoms (such as C-O and C=O due to the loaded dye on the TiO_2_) and adsorbed oxygen species (such as H_2_O or surface hydroxyl group adsorbed from the atmospheric moisture).

The appearance of a peak at around 284.4 eV in both pristine TiO_2_ and SD-TiO_2_ (Figure 3g,h) indicates the presence of C-C and C=C bonds. As we mentioned in the XPS survey spectrum, the presence of the XPS C 1s spectrum of the pristine TiO_2_ in Figure 3g is primarily attributed to atmospheric absorption before the XPS measurements. While, the XPS C 1s spectrum of the SD-TiO_2_ (Figure 3h) as compared the XPS C 1s spectra of pristine TiO_2_ display additional peaks (such as C-O, C=O, and O-C=O) [44,45,46] arising from the bonding of functional groups in the strawberry dye with pristine TiO_2_. These peaks are absent in the XPS spectra of pristine TiO_2_. Moreover, the XPS spectra of the SD-TiO_2_ nanoparticles demonstrate changes in the intensity and position of these peaks, indicating alterations in the electronic structure of the dye molecule upon adsorption onto the TiO_2_ surface. The XPS analysis confirms the presence of dye molecules on the TiO_2_ and successfully verifies the sensitization of TiO_2_ with dye molecules. This sensitization can enhance the photocatalytic and photoelectrochemical properties of the material.

#### 3.1.3. TGA/DTA

To investigate the interaction between pristine TiO_2_ and the strawberry dye, TGA/DTA analysis was conducted. The results of the TGA/DTA analysis for pristine TiO_2_ and SD-TiO_2_ are presented in Figure 4. The TGA plot of pristine TiO_2_, (Figure 4a), exhibited weight loss in two stages within the temperature range of 45–500 °C. The first stage (45–250 °C) was attributed to the loss of adsorbed water on the TiO_2_ surface. The second stage (250–500 °C) was due to the thermal dissociation of the OH group of TiO_2_ and the removal of organic residues or surface-bound molecules that may have originated from the synthesis or handling process [47]. The weight loss recorded was 5.55%, with a residual of 94.45% for TiO_2_. In addition, the DTG curve indicated that the maximum velocity of TiO_2_ dissociation occurred at around 400 °C.

In contrast, the TG analysis of SD-TiO_2_ (Figure 4b) displayed three dissociation stages in the temperature range of 45–520 °C. Our findings clearly indicate that the TGA curve descends until it reaches a plateau at around 520 °C. In Figure 4, the initial slight weight loss up to 162 °C is attributed to the physical desorption of surface-adsorbed water. The subsequent rapid weight loss in two stages, between 162 °C and 373 °C and between 420 °C and 520 °C, respectively, is linked to the thermal decomposition of the dye molecules that cap the TiO_2_ nanoparticles. The overall weight loss observed is 15.92%, with a residual of 84.08%. The DTG curve indicated that the maximum velocity of SD-TiO_2_ dissociation occurred at around 290 °C. The noticeable difference in weight loss between pristine TiO_2_ and SD-TiO_2_ indicated the interaction of the strawberry dye with the TiO_2_ surface.

Overall, the TGA analysis provided compelling evidence of the interaction between the strawberry dye and the TiO_2_ surface, thus confirming the successful sensitization of TiO_2_ nanoparticles with the strawberry dye.

#### 3.1.4. UV-Vis Absorption Spectra

UV-Vis spectroscopy is a valuable technique for investigating the electronic properties of materials and their interactions with other compounds, providing insights applicable to various fields. The UV-Vis spectrum of pristine TiO_2_ as shown in Figure 5 displayed a sharp absorption band in the UV region (300–400 nm). This absorption is attributed to the presence of TiO_2_, which possesses a bandgap energy corresponding to the UV region of the electromagnetic spectrum [47]. In contrast, The UV-Vis spectrum of the strawberry dye (Figure 5) exhibited a wide absorption band spanning 300–700 nm. This absorption band is a result of the presence of conjugated pi-electron systems within the dye molecules, enabling them to absorb light in the visible spectrum [48].

Notably, the UV-Vis spectrum of SD-TiO_2_ (Figure 5) exhibited an increased absorption in the visible region compared to pristine TiO_2_. We can observe that the absorption band of SD-TiO_2_ shows an appreciable red shift towards higher wavelengths in comparison to the absorption band of both pristine TiO_2_ and strawberry dye, as shown in the inset figure. This shift towards lower energy can be attributed to the interaction between the dye molecules and the TiO_2_ surface, which enhances the absorption of visible light. When the sensitizing dye is absorbed onto the TiO_2_ surface, it can absorb light at particular wavelengths, especially in the visible and sometimes UV regions. This absorbed light energy is then utilized to excite electrons within the dye molecule to higher energy levels. Subsequently, these excited electrons can be injected into the conduction band of TiO_2_, thereby enhancing TiO_2_′s efficiency in absorbing and utilizing photons. Such a phenomenon holds promise for applications in photocatalysis and solar energy conversion.

#### 3.1.5. XRD

X-ray diffraction (XRD) can be used to investigate the impact of SB dye on the crystal structure of TiO_2_ nanoparticles. The XRD patterns of pristine TiO_2_ and SD-TiO_2_ are presented in Figure 6a,b, respectively. Pristine TiO_2_ consists of both anatase and rutile phases, as indicated by the observed peaks in Figure 6a. The observed peaks at various angles, including 25.3°, 37.8°, 48.1°, 54.1°, 55.8°, 62.7°, and 68.8°, which correspond to the planes (101), (004), (200), (105), (221), (204), and (116), respectively, can be associated with the anatase phase of TiO_2_ [49]. The peaks at 27.3° (110), 35.9° (101), and 41.3° (111) are attributed to the rutile phase [49].

It is worth noting that the XRD pattern of SD-TiO_2_ in Figure 6b closely resembles that of pristine TiO_2_, indicating that the phases of TiO_2_ remain unchanged following sensitization with dye molecules. However, there may be slight differences in peak positions and intensities due to the interaction between the dye molecules and the TiO_2_ surface. Therefore, the XRD analysis confirms that the strawberry dye does not induce any phase changes in TiO_2_.

#### 3.1.6. SEM

Scanning electron microscopy (SEM) is a high-resolution technique used to examine the morphology and surface topography of materials. Figure 7a,b present the SEM analysis of pristine TiO_2_ and SD-TiO_2_, respectively. Through SEM analysis, valuable insights can be gained regarding the impact of strawberry dye molecules on the surface morphology and topography of TiO_2_, which can be significant for enhancing their photocatalytic and photoelectrochemical properties.

The SEM image of pristine TiO_2_ reveals spherical nanoparticles with a uniform size distribution. However, the SEM image of SD-TiO_2_ may exhibit alterations in surface morphology and topography when compared to pristine TiO_2_ nanoparticles. The presence of dye molecules on the TiO_2_ surface may induce the formation of dense agglomerates consisting of irregular nanoparticles. These agglomerates could enhance the light-absorption and scattering properties of the TiO_2_ nanoparticles, thereby offering benefits for photocatalytic applications.

#### 3.1.7. BET

The BET (Brunauer–Emmett–Teller) analysis is a widely used method for determining the surface area and pore size distribution of materials. Dye molecules can be adsorbed onto adsorbents through physical or chemical processes [50]. Physical adsorption involves van der Waals forces and hydrogen bonding, enabling the dye molecules to adhere to the adsorbent surface. In contrast, chemisorption occurs when the dye molecules or ions establish a chemical connection with specific surface functional groups or sites. In both cases, the adsorption of dye alters the surface parameters of the adsorbents.

In this study, BET analysis was performed on both pristine TiO_2_ nanoparticles and SD-TiO_2_. The N_2_ adsorption–desorption isotherms and BJH (Barrett–Joyner–Halenda) pore size distribution of pristine TiO_2_ nanoparticles and SD-TiO_2_ are presented in Figure 8a,b, respectively. Both isotherms exhibit type IV characteristics, indicating the presence of mesopores [51]. The hysteresis loop of SD-TiO_2_ is less pronounced compared to pristine TiO_2_, suggesting a lower degree of pore interconnectivity in the former sample.

As shown in Table 1, BET analysis revealed that the surface area of pristine TiO_2_ nanoparticles is 50.60 m^2^/g, while SD-TiO_2_ has a surface area of 34.20 m^2^/g. This decrease in surface area for SD-TiO_2_ can be attributed to the adsorption of strawberry dye molecules onto the TiO_2_ surface, which reduces the available surface area for N_2_ adsorption [17]. Additionally, the pore volume of SD-TiO_2_ (0.58 cm^3^/g) is lower than that of pristine TiO_2_ nanoparticles (1.80 cm^3^/g), indicating that the presence of strawberry dye molecules leads to a decrease in the pore volume of TiO_2_ [17]. The BJH method was used to determine the pore size distribution, revealing a pore radius of 28 nm for pristine TiO_2_ nanoparticles and 19 nm for SD-TiO_2_. The considerable reduction in pore radius for SD-TiO_2_ indicates that the pores in SD-TiO_2_ are smaller and more uniform in size due to the presence of strawberry dye molecules.

The results of the BET analysis indicate that the presence of strawberry dye molecules on the TiO_2_ surface significantly impacts its surface properties. These changes are likely to affect the photocatalytic activity of TiO_2_. For instance, the decrease in surface area and pore volume of SD-TiO_2_ compared to pristine TiO_2_ nanoparticles could reduce the number of active sites available for photocatalytic reactions. However, the smaller and more uniform pore size distribution of SD-TiO_2_ has the potential to enhance photocatalytic activity by facilitating the diffusion of reactants and products.

### 3.2. Photocatalytic Activity

We successfully synthesized 4-aryl/heteroaryl-6-(3-coumarinyl) pyrimidin-2 (1H)-ones 4a-h through a multicomponent reaction. This involved combining 3-acetyl coumarin (1), substituted aldehydes (2a–h), and urea (3) using SD-TiO_2_ as a photocatalyst at room temperature (Figure 1). SD-TiO_2_ is an environmentally friendly and cost-effective option. Its use made the experimental process simple, and the photocatalyst could be easily removed through filtration, preventing the generation of harmful acidic waste. By irradiating the reactions with visible light and optimizing the photocatalyst concentration (1 mg/mL of ethanol), we successfully obtained high yields of various 4-aryl-6-(3-coumarinyl) pyrimidin-2(1H)-ones derivatives. The simplicity and ease of product isolation make this process a convenient one-pot procedure. Furthermore, the reactions proceeded faster and yielded higher amounts of the desired product compared to traditional methods.

Our study demonstrates the effectiveness of SD-TiO_2_ as a photocatalyst for the synthesis of coumarin derivatives under visible light irradiation. To investigate further, we conducted experiments on a model reaction to study its behavior under different conditions, such as reaction time, catalyst concentration, and light intensity. We used the ^1^H-NMR and ^13^C-NMR spectra (as shown in the Appendix A) to record the results of these experiments, and then included them in tables. This investigation will aid in optimizing reaction conditions and designing more efficient photocatalytic systems in the future.

Initially, the photocatalytic performance and yield of the reaction were explored by varying the concentration of the photocatalyst while keeping the light intensity fixed. Table 2 displays the results, which indicate that the photocatalytic performance and yield are dependent on the concentration of the photocatalyst. In the absence of any photocatalyst, the yield percentage of the reaction was only 20% after 30 min, indicating poor reaction performance. The yield percentage significantly increased with the presence of the SD-TiO_2_ photocatalyst, reaching 96%, 95%, 92%, 88%, and 82% for 1 mg/mL, 2 mg/mL, 3 mg/mL, and 4 mg/mL of photocatalyst after 30, 60, 90, 120, and 150 min, respectively. These results confirm that the faster reaction rates and higher conversion yields are dependent on the photocatalyst concentrations. Utilizing low concentrations of SD-TiO_2_, such as 1 and 2 mg/mL, was advantageous for improving the photocatalytic performance, leading to increased conversion yields within a shorter reaction time when compared to other concentrations of SD-TiO_2_. The optimal photocatalyst concentration was determined to be 1 mg/mL, resulting in the highest conversion yield within a reasonable reaction time.

To compare the superiority of SD-TiO_2_ over pristine TiO_2_, we conducted a study on the synthesis of a model compound described in Table 3. This study involved the photocatalytic synthesis of coumarin derivatives using either pristine TiO_2_ or SD-TiO_2_ under visible light irradiation. We examined the photocatalytic performance and yield of the reaction with a fixed light intensity and optimum photocatalyst concentration, while varying the irradiation time. The results revealed that both pristine TiO_2_ and SD-TiO_2_ were effective photocatalysts for the synthesis of coumarin derivatives under visible light irradiation. The yield percentage of the reaction increased with longer irradiation times up to a certain point, beyond which, further increases in irradiation time led to decreased performance. The sensitization of TiO_2_ nanoparticles with strawberry dye enhanced the photocatalytic performance, resulting in higher conversion yields in a shorter reaction time compared to pristine TiO_2_.

For example, at 30 min of irradiation time, the yield percentage was 85% for pristine TiO_2_ and 96% for SD-TiO_2_, indicating the superior photocatalytic performance of the latter. Similarly, at 60, 90, 120, and 150 min of irradiation time, the yield percentage consistently favored SD-TiO_2_ over pristine TiO_2_. These findings suggest that the sensitization of TiO_2_ nanoparticles with strawberry dye enables faster reaction rates and higher conversion yields under visible light irradiation.

Overall, the sensitization by the dye can improve the absorption of visible light by the TiO_2_ nanoparticles, thereby enhancing the efficiency of the photocatalytic reaction by increasing the reaction rate and the conversion yields.

In contrast, Table 4 presents a study on the synthesis of coumarin derivatives using SD-TiO_2_ under visible-light irradiation. The focus was on investigating the impact of light intensity on the photocatalytic performance and yield of the reaction, while maintaining a fixed irradiation time and optimum concentration. The results revealed a significant correlation between light intensity and the reaction’s performance. Lower light intensities resulted in lower yield percentages, whereas higher intensities led to higher yields. For instance, at 20 mW cm^−2^, the yield percentage was 68%, gradually increasing to 96% at 100 mW cm^−2^. However, further increases in light intensity caused a slight decrease in the yield percentage.

The sensitivity of the reaction to light intensity can be attributed to the dependence of the reaction rate on the number of photons absorbed by the photocatalyst. When the light intensity is lower, the photocatalyst absorbs fewer photons, resulting in a slower reaction rate and a lower conversion yield. Conversely, higher light intensities lead to increased photon absorption, resulting in a faster reaction rate and a higher conversion yield. This highlights the role of sensitizing TiO_2_ nanoparticles with strawberry dye, which enhances the photocatalytic performance and enables higher conversion yields at higher light intensities.

Table 5 presents a comparison of the photocatalytic and thermocatalytic performances of SD-TiO_2_ nanoparticles for the synthesis of coumarin derivatives under the same operating conditions of fixed light intensity and temperature. The results consistently showed that the photocatalytic performance surpassed the thermocatalytic performance, with higher yields observed at all reaction times. This superior performance in the photocatalytic reaction can be attributed to the enhanced absorption of visible light by the material, resulting in higher efficiency. The modification of TiO_2_ nanoparticle surface properties through the dye also contributes to the observed higher conversion yields in the photocatalytic reaction. On the other hand, the thermocatalytic reaction relies solely on the thermal energy provided to the system, which is less effective in promoting the reaction compared to photocatalytic reactions.

With increasing reaction time, the photocatalytic and thermocatalytic performance becomes more pronounced. For example, at the shortest recorded reaction time (30 min), the yield percentages were 72 and 96% for the thermocatalytic and the photocatalytic reaction, respectively, indicating a relatively small difference in performance. However, at the longest recorded reaction time, the yield percentage of the thermocatalytic reaction decreases significantly, while the photocatalytic reaction consistently maintains a high yield percentage. This study thus demonstrates the superior performance of SD-TiO_2_ in the photocatalytic synthesis of coumarin derivatives under visible light irradiation compared to the thermocatalytic synthesis at a fixed temperature under the same operating conditions.

In Table 6, the synthesis of 4-aryl/heteroaryl-6-(3-coumarinyl)pyrimidin-2(1H)-one derivatives using SD-TiO_2_ as the photocatalyst is explored with various substituted aldehydes. The results indicate that the performance of the photocatalyst is influenced by the specific substituted aldehyde used. Generally, the reaction time and yield were similar for substituted electron-withdrawing and electron-donating aldehydes (compounds **4a**, **4e**). However, the use of heteroaldehydes (entries 6–8, Table 6) resulted in a significant reduction in yield and required longer reaction times. This suggests that the properties of the aldehyde group have a notable impact on the catalyst’s performance, potentially due to differences in their electronic and steric characteristics.

These findings align with prior studies [46] on the synthesis of 4-aryl/heteroaryl-6-(3-coumarinyl)pyrimidin-2(1H)-one derivatives, which have similarly demonstrated the influence of starting materials on catalyst performance. These observations underscore the importance of the meticulous selection of starting materials when designing and optimizing chemical reactions, particularly in the development of novel catalysts. In summary, this study showcases the potential of SD-TiO_2_ as a catalyst for synthesizing 4-aryl/heteroaryl-6-(3-coumarinyl)pyrimidin-2(1H)-one derivatives while shedding light on the factors that can impact catalyst performance. The outcomes of this study can guide the future design of more efficient and selective catalysts for similar reactions.

### 3.3. Photocatalyst Reusability

The evaluation of photocatalyst reusability holds great significance in catalyst design and optimization. The capacity to reuse a catalyst offers notable benefits, such as cost reduction and a decreased environmental impact, thus enhancing sustainability and economic viability. The outcomes of this study yield valuable insights into the reusability potential of SD-TiO_2_ as a photocatalyst, thereby aiding the development of more efficient and environmentally friendly approaches. To evaluate the photocatalyst’s reusability, Table 7 presents an examination of its photocatalytic performance and yield over four consecutive runs in the synthesis of various coumarin derivatives under visible light irradiation. The experiment was conducted with a fixed irradiation time of 30 min and an optimum concentration of 1 mg/mL of SD-TiO_2_ nanoparticles in the reaction mixture.

The study findings indicate that the photocatalyst maintained its photocatalytic performance and yield throughout three consecutive recycling processes. In the initial run, employing 10 mg of fresh photocatalyst resulted in a yield percentage of 96%. Subsequent recycling processes involved the extraction of decreasing amounts of photocatalyst after each run, yet the yield percentage remained relatively high. Specifically, the first recycling process involved a retrieval of 9 mg of the photocatalyst, yielding 95%. In the second recycling process, 8 mg of the photocatalyst was obtained, yielding 93%. Finally, in the third recycling process, 8 mg of the photocatalyst was retrieved, resulting in a yield percentage of 92%.

The findings from this study indicate that the photocatalyst showcased favorable reusability, preserving its effectiveness across multiple runs. The noteworthy yield percentage observed in the subsequent recycling processes suggests that the photocatalyst maintained its catalytic activity, even after the extraction of a portion of the material. These results highlight the promising prospects of SD-TiO_2_ as a durable and effective photocatalyst for synthesizing coumarin derivatives under visible light irradiation. Moreover, its ability to be reused for multiple cycles adds to its appeal and practicality.

### 3.4. Photocatalytic Mechanism

The photocatalytic process involved in the synthesis of coumarin derivatives under visible light irradiation using SD-TiO_2_ or pristine TiO_2_ nanoparticles follows a specific mechanism. It begins with the generation of electron–hole pairs, which then undergo condensation and cyclization reactions, resulting in the formation of the desired coumarin derivatives. The utilization of SD-TiO_2_ broadens the nanoparticles’ absorption range to include visible light, thereby enhancing the overall utilization of light energy. In this photocatalytic reaction, a strawberry dye molecule absorbs visible light, initiating the generation of electron–hole pairs. The excited electrons are subsequently transferred to the surface of the TiO_2_, where they participate in the reduction of 3-acetyl coumarin, substituted aldehyde, and urea [29,32]. To optimize the efficiency of the process, ethanol can be employed as a hole scavenger, effectively removing the holes and preventing their recombination with electrons [54]. Figure 9 illustrates the proposed photocatalytic mechanism for the synthesis of coumarin derivatives under visible light irradiation using SD-TiO_2_, 3-acetyl coumarin, substituted aldehydes (**2a–h**), urea, and ethanol.

Furthermore, the condensation reaction in this process can be divided into two distinct steps, as depicted in Figure 2. Initially, the condensation of the aldehyde and the methyl group of 3-acetyl coumarin forms a stabilized carbenium ion. In the second step, the urea undergoes nucleophilic addition to this intermediate, leading to rapid dehydration and the production of the desired product.

## 4. Conclusions

In summary, our study highlights the promising application of SD-TiO_2_ as a photocatalyst in the synthesis of coumarin derivatives under visible light irradiation. Notably, SD-TiO_2_ exhibited superior photocatalytic activity compared to pristine TiO_2_ nanoparticles, which is attributed to its effective utilization of visible light facilitated by the dye. Through our investigation, we also identified the optimal operating conditions for the photocatalytic synthesis, which can guide future designs of photocatalytic systems targeting coumarin derivative synthesis. The proposed mechanism sheds light on the reaction pathway and intermediates involved in this synthesis process, offering valuable insights. Additionally, the successful reusability test suggests the practical feasibility of SD-TiO_2_ for large-scale photocatalytic synthesis. Altogether, this study makes a valuable contribution to the advancement of sustainable and efficient photocatalytic systems for organic synthesis.

## Data Availability

Data will be provided upon request.

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
