# Peer review of "Photocatalytic Synthesis of Coumarin Derivatives Using Visible-Light-Responsive Strawberry Dye-Sensitized Titanium Dioxide Nanoparticles"

_nanomaterials, 2023, doi:10.3390/nano13233001_

Round 1

Reviewer 1 Report

Comments and Suggestions for Authors

The authors reported the application of strawberry dye-sensitized TiO2 for photocatalytic synthesis of coumarin derivatives. They did a lot of work for synthesis and characterisation of catalysts, photocatatlytic activity evaluation.

1. The XPS spetrum of Ti 2p is missed. It should be showed to check if Ti ions on the surface are reduced during the preparation process.

2. Thermic curve of TGA is missed. The operation condistions of TGA analysis is also missed. And the weight loss of SD-TiO2 from 45-200 degree C is very low, it is not normal because the prepared catalyst was not dried or thermal treatment. The weigh loss from 250 degree C to 600 degree C is about 45%, and the residual 22%, mean that the weigh of dyer is more than 2 times to compare to that of TiO2 and SD covered all the surface of TiO2 in the SEM image. However, the surface of SD-TiO2 is still remained. Maybe treatment sample conditions of the BET analysis was not suitable and the surface of the catalyst was changed. Please provide the treatment conditions. Almost  mesopores of TiO2 was covered by dye and the secondary pores were formed, not as the authors reported: "The considerable reduction in pore radius for SD-TiO2 indicates that the pores in SD-TiO2 are 290 smaller and more uniform in size due to the presence of strawberry dye molecules".

3. The solvent for extraction of SD was ethanol and ethanol was also used as the solvent for the reaction. It is the reason why SD was extracted from catalyst due to the weigh loss of catalyst after reaction.

4. The yield of products at "The yield 317 percentage significantly increased with the presence of SD-TiO2 photocatalyst, reaching 96%, 95%, 92%, 88% and 82% 318 for 1 mg/mL, 2 mg/mL, 3 mg/mL, and 4 mg/mL of photocatalyst after 30, 60, 90, 120 and 150 min, respectively" in table 2 is not suitable. The more catalyst amount was used, the faster reaction rate was. The authors checked the yield of products at longer time with higher catalyst amount, maybe the reaction was overated and the products were converted to by products. 

5. The authors have just reported on the yield of products but they have not mentioned the conversion of raw materials, so we can not checked if the other by-products occured or not.

Comments on the Quality of English Language

Please check and rewrite some sentences.

Author Response

Compliance to reviewer’s and editor’s comments

Journal Nanomaterials (ISSN 2079-4991)

Manuscript ID nanomaterials-2632274

Title: Photocatalytic Synthesis of Coumarin Derivatives using Visible Light Responsive Strawberry Dye-Sensitized Titanium Dioxide Nanoparticles

Authors: Mshari A. Alotaibi, Abdulrahman I. Alharthi, Talal. F.Qahtan, Satam Alotibi,  Md. Afroz Bakht, and Amani M. Alansi

We would like to express our great appreciation to the editor and the reviewers for their feedback to improve our manuscript. We appreciate the fact that our manuscript has been checked rigorously by reviewers with knowledge of the specific subject area. The manuscript has been revised according to the reviewers’ concerns. Attached is a point-by-point response to the constructive comments and suggestions of the reviewers.

Reviewer 2 Report

Comments and Suggestions for Authors

The authors reported dye-sensitized TiO2 (SD-TiO2) in the photocatalytic synthesis of coumarin derivatives under visible light. This work is interesting and suitable for publication after addressing the following issues.

(1) The peak position of rutile (110) is incorrect. Please double check it.

(2) How about the stability of SD-TiO2?

(3) From Figure 8, it is obviously belonged to mesoporous structure. It should be analysized deeply and some related literatures may be helpful (Adv. Funct. Mater. 2011, 21, 1922; J. Am. Chem. Soc. 2014, 136, 9280).

(4) Some grammar errors should be corrected before publication.

Comments on the Quality of English Language

Minor editing of English language required.

Author Response

Compliance to reviewer’s and editor’s comments

Journal Nanomaterials (ISSN 2079-4991)

Manuscript ID nanomaterials-2632274

Title: Photocatalytic Synthesis of Coumarin Derivatives using Visible Light Responsive Strawberry Dye-Sensitized Titanium Dioxide Nanoparticles

Authors: Mshari A. Alotaibi, Abdulrahman I. Alharthi, Talal. F.Qahtan, Satam Alotibi,  Md. Afroz Bakht, and Amani M. Alansi

Response to Reviewer #2

Thank you for your detailed comments and suggestions regarding the manuscript. We appreciate your feedback and will address each of your points below. The writing was further polished as highlighted by the blue fonts in the revised manuscript.

Reviewer 3 Report

Comments and Suggestions for Authors

The studies were well carried over, the detailed discussion was illustrated the introduction section, if author could provide with more bullets points to the specific advantage of this current study would be more attractive to this article reader.

Author Response

Compliance to reviewer’s and editor’s comments

Journal Nanomaterials (ISSN 2079-4991)

Manuscript ID nanomaterials-2632274

Title: Photocatalytic Synthesis of Coumarin Derivatives using Visible Light Responsive Strawberry Dye-Sensitized Titanium Dioxide Nanoparticles

Authors: Mshari A. Alotaibi, Abdulrahman I. Alharthi, Talal. F.Qahtan, Satam Alotibi,  Md. Afroz Bakht, and Amani M. Alansi

Response to Reviewer #3

Thank you for your detailed comments and suggestions regarding the manuscript. We appreciate your feedback and will address each of your points below. The writing was further polished as highlighted by the blue fonts in the revised manuscript.

Reviewer 4 Report

Comments and Suggestions for Authors

1. The manuscript must be edited by a senior author or an English professional to improve English level.

2. The abstract needs improvement. Please add more information on research results obtained from the characterization of the materials. It should be informative and completely self-explanatory.

3. The novelty of the research should be clearly described at the end of the introduction part.

4. Further photochemical analyses like photoluminescence spectroscopy and photocurrent responses should be performed to investigate the improvements in the differences between the optical properties of TiO2 and PAEs-TiO2.

5. Line 236: While the authors suggest the presence of a red shift in the absorption band of SD-TiO2 compared to pristine TiO2, it appears that there is no significant red shift. Instead, it seems that the baseline of the SD-TiO2 spectrum is slightly elevated compared to pristine TiO2. It's important to accurately describe the observed changes in the UV-Vis spectrum, as this will impact the interpretation of the results.

6. Further photochemical analyses like photoluminescence spectroscopy, photocurrent responses, and EIS should be performed to investigate the improvements in the differences between the optical properties of pristine and modified TiO2.

7. Some references (especially those in introduction part) should be replaced with recently published papers including:

ACS Applied Materials & Interfaces, 15 (2023) 27277-27284. 10.1021/acsami.3c02340

International Journal of Hydrogen Energy 2019, 44 (44), 24162-24173. DOI: https://doi.org/10.1016/j.ijhydene.2019.07.129.

Separation and Purification Technology, 325 (2023) 124706. https://doi.org/10.1016/j.seppur.2023.124706

RSC Advances, 13 (2023) 26484-26508. 10.1039/D3RA05098J

Journal of Cleaner Production 366 (2022), 132761. https://doi.org/https://doi.org/10.1016/j.jclepro.2022.132761.

8. The quality of SEM images should be enhanced, and it is recommended to include images with higher magnifications. Presenting high-quality SEM and HRTEM images would provide additional insights regarding the morphological and structural properties of the synthesized materials.

9. Elemental composition of the samples should be studied quantitatively.

10. The results and discussion section should be improved.  It should be presented with clarity and precision, should be explained by referring to the literature, and should interpret the findings in view of the obtained results.

11. To assess the stability of photocatalysts, it is recommended to conduct successive photocatalysis cycles, followed by elemental composition and FTIR analyses after degradation/adsorption processes.

Comments on the Quality of English Language

Moderate editing of English language required.

Author Response

Compliance to reviewer’s and editor’s comments

Journal Nanomaterials (ISSN 2079-4991)

Manuscript ID nanomaterials-2632274

Title: Photocatalytic Synthesis of Coumarin Derivatives using Visible Light Responsive Strawberry Dye-Sensitized Titanium Dioxide Nanoparticles

Authors: Mshari A. Alotaibi, Abdulrahman I. Alharthi, Talal. F.Qahtan, Satam Alotibi,  Md. Afroz Bakht, and Amani M. Alansi

Response to Reviewer #4

Thank you for your detailed comments and suggestions regarding the manuscript. We appreciate your feedback and will address each of your points below. The writing was further polished as highlighted by the blue fonts in the revised manuscript.

Reviewer 5 Report

Comments and Suggestions for Authors

1. The authors of this manuscript report a specific method to fabricate titanium dioxide nanoparticles modified with strawberry and its use in the photocatalytic synthesis of coumarin derivatives. A range of appropriate instrumental methods (FT-IR, XRD, XPS, TGA/DTA, SEM, SAED) were used for characterization to collect useful pieces of information.

2. Results summarized in Table 6 (8 examples) show excellent yields in 35-60 min treatment despite decreasing catalyst quantity.

3. Reusability

According to data (Table 7) the catalyst provides high yields. It is noted, however, that consistent high yields in a few repeated runs (in this case: four runs) are not a satisfactory measure to claim stability – it is simple a hint. Reusing 10 times would be more convincing.

4. Corrections

i) minor changes

Coumarin in text should not be capitalized; the same correction in lines 101/102 (4-Chlorobenzaldehyde, etc); FTIR vs FT-IR (p. 6, line 135, 183); line 222: TGA/DTA; lines 327+403: 3-Acetylcoumarin (1 mmol), 4-chlorobenzaldehyde; line 401: “…4-aryl/hetroaryl-6-(3-coumarinyl) pyrimidin-2(1H)-one derivatives” – not a full list! Make a careful re-reading of your manuscript!

ii) section 2.4: “To prepare the…dissolve..” etc.

The correct style: “Coumarin derivatives as shown in Scheme 1 were prepared by dissolving 0.5 mmol of 3-acetyl coumarin (1) …etc etc in 10 mL of ethanol in a 50 mL glass beaker.”” The whole text here should be revised accordingly!

iii) List of references –it is a mess and as such it is unacceptable! Missing author names, journal names, etc. etc. Full update-re-work are needed!

See Authors Guide and make changes accordingly! What does “n.d.” stend for? not detemined?

Comments on the Quality of English Language

see above!

Author Response

Compliance to reviewer’s and editor’s comments

Journal Nanomaterials (ISSN 2079-4991)

Manuscript ID nanomaterials-2632274

Title: Photocatalytic Synthesis of Coumarin Derivatives using Visible Light Responsive Strawberry Dye-Sensitized Titanium Dioxide Nanoparticles

Authors: Mshari A. Alotaibi, Abdulrahman I. Alharthi, Talal. F.Qahtan, Satam Alotibi,  Md. Afroz Bakht, and Amani M. Alansi

Response to Reviewer #5

Thank you for your detailed comments and suggestions regarding the manuscript. We appreciate your feedback and will address each of your points below. The writing was further polished as highlighted by the blue fonts in the revised manuscript.

Round 2

Reviewer 1 Report

Comments and Suggestions for Authors

The manuscript is good enough for publication.

Author Response

Thank you for recommending our manuscript for publication. We appreciate your 
positive evaluation.

Reviewer 3 Report

Comments and Suggestions for Authors

It can be accepted.

Author Response

Thank you for recommending acceptance of our manuscript. We appreciate 
your positive evaluation.

Reviewer 4 Report

Comments and Suggestions for Authors

The authors have made some revisions in response to my previous comments, but I still have some significant remaining concerns.

  1. In my earlier report, specifically in comment #5, I pointed out the following issue: "While the authors suggest the presence of a red shift in the absorption band of SD-TiO2 compared to pristine TiO2, it appears that there is no significant red shift. Instead, it seems that the baseline of the SD-TiO2 spectrum is slightly elevated compared to pristine TiO2. Accurate description of the observed changes in the UV-Vis spectrum is crucial, as it directly impacts the interpretation of the results." The authors have attempted to address this concern by explaining, "When the sensitizing dye is absorbed onto the TiO2 surface, it can absorb light at specific wavelengths, especially in the visible and sometimes UV regions. This absorbed light energy is then utilized to excite electrons within the dye molecule to higher energy levels. Subsequently, these excited electrons can be injected into the conduction band of TiO2, thereby enhancing TiO2's efficiency in absorbing and utilizing photons." However, I still find this explanation unacceptable as there is no significant absorption in visible light, and the baseline of the SD-TiO2 spectrum remains slightly elevated compared to pristine TiO2. Furthermore, this makes comment #4 more critical, which suggests that further photochemical analyses such as photoluminescence spectroscopy and photocurrent responses should be performed to investigate the differences in the optical properties of TiO2 and PAEs-TiO2.
  2. Several references, especially in the introduction section, should be updated to include more recent papers, such as:
    • ACS Applied Materials & Interfaces, 15 (2023) 27277-27284. DOI: 10.1021/acsami.3c02340
    • International Journal of Hydrogen Energy, 2019, 44 (44), 24162-24173. DOI: https://doi.org/10.1016/j.ijhydene.2019.07.129
    • Separation and Purification Technology, 325 (2023) 124706. DOI: https://doi.org/10.1016/j.seppur.2023.124706
    • RSC Advances, 13 (2023) 26484-26508. DOI: 10.1039/D3RA05098J
    • Journal of Cleaner Production, 366 (2022), 132761. DOI: https://doi.org/10.1016/j.jclepro.2022.132761

Without a proper resolution to these concerns, I cannot recommend the manuscript for acceptance.

Comments on the Quality of English Language

Minor editing of English language required

Reviewer 5 Report

Comments and Suggestions for Authors

The only exception is the reference list!

- ref. 4: missing information African J. Pure Applied Chem. 4, 74-86.

- ref. 13: authors are missing: Sharma D, Dhayalan V, Manikandan C, Dandela R

- ref. 25: RSC Adv., 2015,5, 31415-31421.

- ref. 27: Photosensitization effect on visible-light-induced photocatalytic performance of chlorophyll and flavonoid nanostructures: kinetic and isotherm studies. Bull. Mater. Sci. 2019, 42, 248.

- ref. 33: author names are missing! Food Funct. 2014, 5, 1939–1948.

- ref. 40: 2011, 13, 113

Use offcial journal ebbreviations! This is the source: https://cassi.cas.org/search.jsp

Author Response

Thank you for recommending acceptance of our manuscript. We appreciate 
your positive evaluation. We apologize for the oversight, and we have 
diligently addressed and rectified all the errors that you pointed out in the
references.

Round 3

Reviewer 3 Report

Comments and Suggestions for Authors

The current form of the manuscript can be accepted for publication 

Reviewer 5 Report

Comments and Suggestions for Authors

The authors have made satisfactory changes/corrections as indicated in my evaluation including the reference list!